# Viral Entry Inhibitors Protect against SARS-CoV-2-Induced Neurite Shortening in Differentiated SH-SY5Y Cells

**DOI:** 10.3390/v15102020

**Published:** 2023-09-28

**Authors:** Margaux Mignolet, Jacques Gilloteaux, Nicolas Halloin, Matthieu Gueibe, Kévin Willemart, Kathleen De Swert, Valéry Bielarz, Valérie Suain, Ievgenia Pastushenko, Nicolas Albert Gillet, Charles Nicaise

**Affiliations:** 1URPhyM, NARILIS, Faculté de Médecine, Université de Namur, Rue de Bruxelles 61, 5000 Namur, Belgium; margaux.mignolet@unamur.be (M.M.); jgilloteaux@sgu.edu (J.G.); nicolas.halloin@unamur.be (N.H.); matthieu.gueibe@student.unamur.be (M.G.); kathleen.deswert@unamur.be (K.D.S.); valery.bielarz@unamur.be (V.B.); 2Department of Anatomical Sciences, St George’s University School of Medicine, Newcastle upon Tyne NE1 JG8, UK; 3URVI, NARILIS, Faculté des Sciences, Université de Namur, Rue de Bruxelles 61, 5000 Namur, Belgium; kevin.willemart@unamur.be (K.W.); nicolas.gillet@unamur.be (N.A.G.); 4Laboratoire d’Histologie Générale, Faculté de Médecine, Université Libre de Bruxelles, Route de Lennik 808, 1070 Bruxelles, Belgium; valerie.suain@ulb.be (V.S.); ievgenia.pastushenko@ulb.be (I.P.)

**Keywords:** SARS-CoV-2, SH-SY5Y, neurotropism, cytopathic effect, entry inhibitor

## Abstract

The utility of human neuroblastoma cell lines as in vitro model to study neuro-invasiveness and neuro-virulence of SARS-CoV-2 has been demonstrated by our laboratory and others. The aim of this report is to further characterize the associated cellular responses caused by a pre-alpha SARS-CoV-2 strain on differentiated SH-SY5Y and to prevent its cytopathic effect by using a set of entry inhibitors. The susceptibility of SH-SY5Y to SARS-CoV-2 was confirmed at high multiplicity-of-infection, without viral replication or release. Infection caused a reduction in the length of neuritic processes, occurrence of plasma membrane blebs, cell clustering, and changes in lipid droplets electron density. No changes in the expression of cytoskeletal proteins, such as tubulins or tau, could explain neurite shortening. To counteract the toxic effect on neurites, entry inhibitors targeting TMPRSS2, ACE2, NRP1 receptors, and Spike RBD were co-incubated with the viral inoculum. The neurite shortening could be prevented by the highest concentration of camostat mesylate, anti-RBD antibody, and NRP1 inhibitor, but not by soluble ACE2. According to the degree of entry inhibition, the average amount of intracellular viral RNA was negatively correlated to neurite length. This study demonstrated that targeting specific SARS-CoV-2 host receptors could reverse its neurocytopathic effect on SH-SY5Y.

## 1. Introduction

Although considered as a respiratory pathogen, SARS-CoV-2 (severe acute respiratory syndrome coronavirus 2), the virus that caused the COVID-19 pandemic, is also responsible in a substantial number of cases of extrapulmonary manifestations [1]. Among them, acute neurological manifestations span from the mildest to the most severe symptomatology: dysosmia, dysgeusia, headache, confusion, Guillain–Barré syndrome, myelitis, encephalitis, cerebrovascular events, or acute hemorrhagic necrotizing encephalopathy [2]. In acute patients with severe illness, a large longitudinal fMRI study has revealed that SARS-CoV-2 impacts both brain morphology and functions: a reduction in grey matter thickness in the orbito-frontal cortex and parahippocampal gyrus, a global reduction in brain size, and a significant cognitive decline [3,4]. Longitudinal PET imaging study has also evidenced decreased metabolic activity in multiple brain regions of such patients [5]. Even after the resolution of the infection, a minority of patients manifest persistent neurologic symptoms such as ‘brain fog’ during the so-called long COVID or post-acute COVID-19 syndrome [6]. Nowadays, the neurotropism of SARS-CoV-2 has been firmly demonstrated in post-mortem brain tissues from severely affected COVID-19 patients [7,8], as well as in experimentally infected animal models [8,9,10] or in human 2D/3D cell culture systems [11,12,13,14,15,16]. While 2D cultures were mostly utilized to investigate the infectivity of pure populations of human stem cell-derived neurons [16,17], 3D organoids allowed researchers to discover in a cellularly heterogenous model that astrocytes or choroid epithelial cells were also targeted by the virus and could play a role during the course of a brain infection [11,12].

Independent research groups have shown in vitro that several neuronal populations support the entry of SARS-CoV-2, followed or not by genome replication, production, and release of virions [18]. For instance, human iPSC-derived neurons, grown either as neurospheres or brain organoids, were demonstrated as being susceptible to SARS-CoV-2 infection [8,13,14,15,16,17,19,20]. Hence, infected neurons suffered from variable cytopathic effects (CPE), according to the duration of infection or multiplicity of infection (MOI), or even depending on the SARS-CoV-2 variant of concern [21]. The panel of CPE included changes in cell morphology, synaptic loss, neurite shortening, disturbances of energy metabolism, and syncytium formation, up to neuronal death (reviewed in [22,23]). In addition, transcriptional changes were observed, especially in pathways linked to the antiviral or inflammatory response. Post-translational modifications of cytoskeletal proteins and modifications in the epigenome were reported as well [22,23].

To enter the host neurons, SARS-CoV-2 requires the interaction between its Spike (S) protein and cell membrane receptors [24,25,26,27]. This Spike protein is a glycoprotein containing two subunits called S1 and S2. S1 contains an N-terminal domain, a receptor-binding domain (RBD) and two C-terminal domains. S2, the transmembrane subunit, contains the fusion peptide [25]. There is a polybasic cleavage site at the S1/S2 boundary, containing a sequence enriched in arginine residues. The N-terminal domain can recognize glycans or specific protein receptors upon attachment onto the cell surface, while the RBD interacts with Angiotensin-Converting Enzyme-2 (ACE2) receptor, expressed on various human target cells. As such, SARS-CoV-2 hijacks natural gateways, among which are physiologically expressed entry receptors such as ACE2 receptor and Neuropilin-1 (NRP1) coreceptor [28,29]. Two cleavages of the S protein precede the fusion between SARS-CoV-2 membrane and cell host membrane. Those cleavages, the first one at the junction of the S1 and S2 subunits and the second one at the S2′ site, are mediated either by the transmembrane protease serine 2 (TMPRSS2) at the plasma membrane or by cathepsin L in the endolysosome [25], although the second option seems to predominate in cultured neurons [30]. Despite extensive knowledge on the viral entry at the cellular level, the preferential route of access into the central nervous system remains a subject of debate, as well as the individual susceptibility for developing COVID-19-related neurological complications.

Modelling, while understanding SARS-CoV-2 infection and CPE on neurons, is a prerequisite in the establishment of large drug-screening platforms that characterize the effects of antiviral compounds, critically needed in the therapeutical arsenal against the neurological complications. This is where immortalized neuronal cell lines can serve as convenient and low-cost cell culture system for testing drugs that interfere with viral entry, replication, or release. We and others have demonstrated that SH-SY5Y neuroblastoma cell line could be susceptible to SARS-CoV-2 infection at various MOIs [24,31,32]. This susceptibility is supported by a significant expression of the ACE2 receptor, as well as the expression of TMPRSS2 protease and NRP1 co-receptor [24,26,31,32]. Our main findings were that SARS-CoV-2 pre-alpha strain was able to invade, at high MOI, retinoic acid-differentiated SH-SY5Y cells, without exerting major CPE-related changes and without inducing massive cell death [31]. In the present report, we sought to exploit the susceptibility of SH-SY5Y cells towards SARS-CoV-2 to extend the characterization of the CPEs on neuron-like cells and preventing them by applying entry inhibitors targeting the ACE2 receptor, the receptor-binding domain (RBD) of the Spike protein, the protease activity of TMPRSS2, and the NRP1 co-receptor.

## 2. Materials and Methods

### 2.1. Cell Culture and Reagents

The SH-SY5Y neuroblastoma (Sigma-Aldrich, 94030304, lot n°21D004, Overijse, Belgium) and VeroE6 cell lines were expanded in DMEM/F12 (Life Technologies, Grand Island, NY, USA) supplemented with 10% fetal bovine serum (Sigma-Aldrich, Overijse, Belgium) and penicillin/streptomycin (Sigma-Aldrich, Overijse, Belgium) under sterile conditions at 37 °C, 95% air humidity, and 5% CO2 concentration. Culture medium was changed twice a week, and cells were passaged at 80–90% confluency. For passage, cells were trypsinized using 0.25% trypsin, 0.53 mM EDTA solution (Sigma-Aldrich, Overijse, Belgium) for 5 min, washed with medium and centrifuged at 130× *g* for 5 min at room temperature. For differentiation experiment, SH-SY5Y cells were seeded at a density of 20,000 cells per cm^2^, on supports pre-coated with 0,1 mg/mL Poly-L-Lysine (Sigma-Aldrich, Overijse, Belgium). To induce neuronal differentiation of SH-SY5Y cells, 10 µM retinoic acid (Sigma-Aldrich, Overijse, Belgium) was added in a low-serum medium (DMEM/F12 supplemented with 1% fetal bovine serum and penicillin/streptomycin). This differentiation was initiated 24 h after seeding and fresh medium was replaced every day for seven days. For entry inhibitors assay, cells were differentiated as above in 24-well plate. After 7 days of differentiation, the supernatant was replaced by 400 μL of fresh medium containing inhibitors 2 h prior the infection. The different concentrations of entry inhibitors were as follows: camostat mesylate 0.1 µM and 10 µM (TOCRIS, Bristol, UK); anti-SARS-CoV-2 RBD neutralizing antibody 0.01 µg/mL and 1 µg/mL (Acro Biosystems, Newark, NJ, USA); human recombinant soluble ACE2 0.1 µg/mL and 10 µg/mL (BioLegend, San Diego, CA, USA); EG00229 1 µM and 100 µM (TOCRIS, Bristol, VA, USA). In mock wells and wells without inhibitor, only 400 μL of vehicle solution was added. After 24 h of infection by SARS-CoV-2, supernatant was harvested, cells were then washed 3 times with PBS and finally lysed for RNA extraction. To mimic inflammatory conditions, TNF-α (Bio-techne, Minneapolis, MN, USA) was added at a concentration of 10 ng/mL 1 h before the infection step. To induce accumulation of lipid droplets, differentiated SH-SY5Y were incubated with 10 µM DAPT (Sigma-Aldrich, Overijse, Belgium) for 5 additional days. At the end of the infection, culture medium was carefully discarded, and cells were washed three times with PBS.

### 2.2. Amplification of Belgian SARS-CoV-2 Strain and Infection Paradigms

Handling of virus stock, including amplification and infection procedures, was performed in a biosafety level-3 laboratory (Université de Namur, Namur, Belgium). The pre-α SARS-CoV-2 isolate was kindly provided by Prof. Piet Maes (KULeuven, Leuven, Belgium). The viral titer was determined using a 50% Tissue Culture Infectious Dose (TCID50) assay. Viral samples were serially 10-fold diluted in DMEM/F12 supplemented with 2% FBS and penicillin/streptomycin. Six replicate dilution series per sample were dispensed on VeroE6 plated at a density of 40,000 cells into wells of a 96-well plate. Along the 5 days incubation at 37 °C and 5% CO2, the occurrence of lysis plaques in the VeroE6 cell layer was monitored for each viral dilution. On day 5, the dilution at which at least 50% of the wells show lysis plaques was used to calculate the TCID50 of the viral stock using the Reed–Muench equation. Unless specified, each experimental design included a MOI 5.0 condition and a MOCK condition, the latter corresponding to culture supernatant from uninfected VeroE6 cells.

SH-SY5Y cells were plated at a density of 100,000 cells per well in 12-well plates. They were either plated directly in the wells for gene/protein expression studies or on glass coverslips for microscopy read-outs. For extracellular detection of SARS-CoV-2 viral RNA, SH-SY5Y cells were plated at a density of 100,000 cells per well in a 6-well plate. VeroE6 cells were plated at a density of 75,000 cells per well in a 6-well plate and were used as positive control. Cells were infected during 2 h using an MOI of 1.0 in low-serum medium. After 2 h, culture medium was discarded for all conditions, and cells were washed three times with PBS. Fresh medium was added to plates. At 2, 24, 48, or 72 h, 100 μL of supernatant was sampled and stored at −80 °C.

### 2.3. RNA Extraction and RT-qPCR

Cells or supernatants were resuspended in 1 mL of TriZol reagent (Life Technologies, Bleiswijk, The Netherlands). Total RNAs were isolated according to manufacturer’s instructions. RNA yield and purity were determined using a spectrophotometer NanoDrop 1000 (Thermo Scientific, Bleiswijk, The Netherlands). Total RNAs were reverse transcribed using the Super Script III Reverse Transcriptase kit according to manufacturer’s instructions (Invitrogen, Merelbeke, Belgium). cDNA samples were used to amplify SARS-CoV-2 gene E with Takyon TaqMan kit [33] and human genes of interest with Takyon SYBR Green kit (Eurogentec, Liège, Belgium) in a Light Cycler 96 device (Roche Diagnostics, Mannheim, Germany). Primer sequences (Eurogentec, Liège, Belgium) are listed in Table 1. The relative gene expression was calculated using the ΔCq method with human *hprt* as housekeeping gene.

### 2.4. Protein Extraction and Resolution

Cells were lysed with RIPA lysis buffer containing protease inhibitor and phosphatase inhibitor cocktail (Sigma-Aldrich, Overijse, Belgium). Samples were sonicated at 20 kHz for 10 s and then centrifuged for 5 min at 13,000× *g* at 4 °C. The supernatant was collected, and protein concentration was determined using a Pierce BCA Protein Assay Kit (ThermoFisher Scientific, Rockford, IL, USA). The samples were stored at −20 °C. Then, 50 µg of protein samples was denatured via boiling for 5 min and loaded onto SDS-PAGE for electrophoresis at 120 V. The gel was then transferred onto a PVDF membrane (ThermoFisher Scientific, Rockford, IL, USA) and then saturated with TBS-0.1% Tween + non-fat dried milk 5% for 1 h at room temperature. The membrane was incubated with the primary antibody overnight at 4 °C (listed in Table 2). The next day, the membrane was washed 3 times with TBS-0.1% Tween (Fisher bioreagents) and then incubated with secondary antibody (listed in Table 3) for 1 h at ambient temperature. All antibodies were diluted in TBS-0.1% Tween + non-fat dried milk 5%. Prior to revelation, the membrane was rinsed 3 times with TBS-0.1% Tween and then incubated for 1 min in a chemiluminescent revelation solution (BM Chemiluminescence Blotting Substrate, Roche Diagnostics, Mannheim, Germany). The reading was performed using an ImageQuant LAS 4000mini system (GE Healthcare, Little Chalfont, UK).

### 2.5. Cell Viability Assays

Lactate dehydrogenase (LDH) activity was measured using an LDH assay kit (CyQUANT LDH cytotoxicity Assay) according to the manufacturer’s instructions (Thermo Fisher Scientific, Waltham, MA, USA). In a 96-well plate, 50 µL of infected SH-SY5Y supernatant was added followed by 50 µL of substrate. The plate was then incubated at room temperature for 30 min protected from the light and was read using a VersaMax spectrophotometer (Molecular Devices, San Jose, CA, USA) at 490 nm and 680 nm. The LDH activity was determined by subtracting the 680 nm absorbance value (background) from the 490 nm absorbance. For MTT metabolic assay, the cells were incubated with 0.5 mg/mL MTT (VWR/Amresco, Solon, OH, USA). Following a 1 h incubation at 37 °C, the supernatant was removed from the plates and isopropyl alcohol was added. The absorbance of MTT was measured using a spectrophotometer (Multiskan Ascent 354 microplate reader; Labsystems, Vantaa, Finland) at 540 nm.

### 2.6. Neuropeptide Y Assay

Neuropeptide Y (NPY) was measured using a commercially available ELISA kit (EZHNPY-25K, Millipore, Temecula, CA, USA), according to the manufacturer’s instructions. Briefly, this assay is a sandwich ELISA based on capture of NPY by human NPY IgG precoated on wells. Thereafter, a secondary biotinylated antibody is binding to NPY after brief washing, followed by conjugation of horseradish peroxidase to the immobilized biotinylated antibodies after further washing. Finally, 3,3′,5,5′-tetra-methylbenzidine substrate is added before measurement of absorbance at 450 nm, corrected from the absorbency at 590 nm (VersaMax spectrophotometer, Molecular Devices, San Jose, CA, USA).

### 2.7. Fluorescent In Situ Hybridization/(Immuno)Staining

Cells were chemically fixed in cold 4% buffered paraformaldehyde for 20 min and then washed with PBS and stored at 4°. Fixed coverslips were processed within one-week post-fixation. The procedure for detecting in situ SARS-CoV-2 gene S mRNA followed manufacturer’s kit instructions (RNAScope, ACDBio, Abingdon, UK) and according to previously published protocols [31,34]. Two specific probes were used to target, respectively, positive or negative single-strand SARS-CoV-2 gene S mRNA (RNAScope Probe-V-nCoV2019-S or RNAScope V-nCoV2019-S-sense, ACDBio, Abingdon, UK). Briefly, cells were permeabilized with PBS + 0,1% Tween 20 for 10 min. After washing, cells were incubated with a solution of protease III (RNAscope Protease III, Advanced Cell Diagnostics Inc, Hayward, CA, USA) for 10 min and subsequently washed. Cells were then incubated with the probe for 2 h at 40 °C. Slides were washed with the RNAscope wash buffer. The amplification step was performed three times, and the enzyme horseradish peroxidase (HRP), TSA Plus Cyanine 3, and HRP blocker were added on slides. Between each step, cells were washed with RNAScope wash buffer. The slides were then incubated with PBS Tween 0.1% + Goat serum 5% for 45 min. Then, cells were incubated overnight with primary antibody and PBS Tween 0.1% + goat serum 5%: monoclonal anti-α-tubulin or anti-ß3-tubulin antibodies (Table 2). The next day, slides were rinsed with PBS 1× for 5 min. Cells were incubated with the secondary antibody Alexa fluor 488 goat anti-mouse IgG (A11001, Invitrogen) for 1 h. Alternatively, intracellular lipid droplets were stained according to a standard protocol using 0.1% Oil Red O solution for 30 min at RT [35]. Cell nuclei were counterstained with DAPI (ProLongTM Gold Antifade reagent with DAPI, Invitrogen, Merelbeke, Belgium). Cells were finally rinsed with PBS for 5 min and kept at 4 °C until imaging. Slides were imaged using an Olympus BX63 epifluorescence microscope equipped with XM10 camera (Olympus Corporation, Tokyo, Japan).

### 2.8. Immunofluorescence

Cells were chemically fixed in cold 4% buffered paraformaldehyde for 20 min and then washed with PBS and stored at 4°. Cells were permeabilized with TBS 1× 1% triton for 10 to 15 min. After washes, cells were saturated with a solution of TBS-Triton 1%, Goat serum, and TBS 1× for 1 h. Following washing, cells were then incubated with the primary antibodies overnight at 4 °C (Table 2). The next day, the slides were rinsed with TBS 1×. Cells were incubated with the secondary antibody (Table 3) for 1 h. After washing, cell nuclei were counterstained with DAPI. Slides were kept at 4 °C until imaging. Slides were imaged using an Olympus BX63 epifluorescence microscope equipped with XM10 camera (Olympus Corporation, Tokyo, Japan).

### 2.9. Electron Microscopy

Cells were chemically fixed for 20 min with glutaraldehyde 4% in Millonig buffer 0.1 M pH 7.4 for 30 min RT and rinsed with Millonig buffer 0.1 M 0.54% glucose for transmission electron microscopy (TEM) or with 2.5% glutaraldehyde in 0.1 M cacodylate buffer for 15 min RT for scanning electron microscopy (SEM). For SEM protocol, coverslips were washed 2 times with cacodylate buffer and stored at 4 °C. After 3 washings with milli-Q water, samples were immersed in 1% osmium tetroxide (OsO4) for 1 h, washed with milli-Q water and then incubated for 20 min in 0.1 M aqueous saturated thiocarbohydrazide. This process (OsO4 followed by thiocarbohydrazide) was repeated twice. After dehydration by incubation in increasing concentrations of ethanol (30°, 50°, 70°, 90°, 100°), samples were dried at the critical point by means of liquid carbon dioxide as a transition fluid, then mounted on a silver-coated supports. The imaging was performed with a scanning electron microscope at 3 kV (Jeol 7500-F). For TEM protocol, coverslips were washed in Millonig’s buffer containing 0.54% sucrose and stored at 4 °C. Cells were then post-fixed in OsO4 2%, dehydrated in ethanol and embedded in epoxy resin LX112, as previously described [36]. Obtained with a diamond knife, ultrathin sections (40–70 nm) were collected on 200 and 300 mesh nickel grids (Micro to Nano, Haarlem, The Netherlands) and contrasted with uranyl acetate and lead citrate. Afterwards, a carbon film was evaporated over the grids. Grids were observed with a Philips Tecnai 10 electron microscope, equipped with a digitized Olympus ITEM platform MegaView G2 image analysis, at an accelerating voltage of 60–80 kV.

### 2.10. Image Analysis

For the measurements of neurite length, the neurite extensions were traced and analyzed using ImageJ software v1.53 [37]. For each experimental condition, at least 100 neuritic processes were measured by two independent investigators in a blind manner (MM and MG). For the measurements of lipid droplet density, the gray value of each lipid droplet was quantified using FIJI software, normalized to the background (blank epon resin) and organelle size, and expressed relative to a 100% density of the nickel TEM grid. For each experimental condition, at least 50 lipid droplets were taken at the same fine structure magnification and quantified by two investigators in a blind manner (NH and JG).

### 2.11. Statistics

Unless specified, all results were expressed as mean values ± standard error of the mean (SEM). For the experiments involving the comparison of two conditions (e.g., mock vs. MOI 5.0), the statistical significance was assessed using an unpaired *t* test. For the experiments involving the comparison of more than two conditions (e.g., different concentrations of entry inhibitors), the statistical significance was assessed using one-way ANOVA followed by Dunnett’s comparison test or Bonferroni method. For the experiments involving the comparison of more than two conditions and a time effect (e.g., mock vs. different MOIs at different timings), the statistical significance was assessed using Two-way ANOVA followed by Tukey’s multiple comparison test. To analyze the degree of relationship between two variables, e.g., the expression of SARS-CoV-2 E gene related to the RdRp gene or related to the neurite length, a Spearman correlation test was conducted. The level of significance was set at *p* < 0.05. All statistical analysis were run on GraphPad Prism (GraphPad Software 9, La Jolla, CA, USA).

## 3. Results

### 3.1. Differentiated SH-SY5Y Were Susceptible, but Not Permissive, to SARS-CoV-2

In accordance with our first intention, the susceptibility of differentiated SH-SY5Y cells towards SARS-CoV-2 infection has been validated, as previously demonstrated by us and others [24,26,31,32,38]. Viral RNA was evidenced using in situ hybridization against the positive strand RNA sequence coding the S gene. There, signals of the fluorescent V-nCoV2019-S probe were mostly found in the perikaryon of infected cells at 24 h post-infection with MOI 5.0 (Figure 1A), and from time to time along neuritic processes (Figure 1K, arrows). In mock condition, the cells were devoid of detectable fluorescence, while confocal analysis of the infected samples confirmed the intracellular localization of S gene mRNA, as observed by the overlapping of the probe signal with the cytoskeletal protein b3-tubulin on transverse optical slice through the perikaryon (Figure 1B). The susceptibility of SH-SY5Y cells was consistent with the significant expressions of known SARS-CoV-2 entry receptors, among those ACE2, TMPRSS2, NRP1, or even CD147 were detected (Appendix A). Following 24 h of SARS-CoV-2 infection, no statistical change in entry receptor mRNAs levels was measured (one-way ANOVA; ACE2, 0.23 ± 0.06 vs. 0.39 ± 0.20; TMPRSS2, 7.50 ± 0.96 vs. 8.5 ± 2.1; NRP1, 5.3 ± 2.5 vs. 3.1 ± 2.2; CD147, 14.0 ± 1.9 vs. 16 ± 0.78; *p* > 0.05 for all receptors between mock and MOI 5.0 conditions).

Differentiated SH-SY5Y cells were also exposed to SARS-CoV-2 at graded MOIs for either 2 or 24 h duration, with a read-out 24 h after infection. The amount of internalized viral particles in infected SH-SY5Y was quantified using a qRT-PCR targeting the SARS-CoV-2 *gene E* (Figure 1C). In SH-SY5Y, a 24 h infection with MOIs 1.0 and 5.0 resulted a significant detection of SARS-CoV-2 RNA from lyzed cells in a dose-dependent manner (two-way ANOVA, MOI effect: F (2, 29) = 168.5, *p* < 0.0001; Tukey’s test, mock vs. MOI 1.0, *p* = 0.024, MOI 1.0 vs. MOI 5.0, *p* < 0.0001). Unlike a 24 h incubation, a 2 h infection resulted in a significant detection of SARS-CoV-2 RNA only at the highest MOI (two-way ANOVA, MOI effect: F (2, 29) = 168.5, *p* < 0.0001; Tukey’s test, mock vs. MOI 1.0, *p* > 0.05, MOI 1.0 vs. MOI 5.0, *p* < 0.0001). It is noteworthy that before the cells were lyzed to isolate their RNA content (viral + human total RNAs), the cells were thoroughly washed three times with PBS to avoid the detection of remnant extracellular inoculum. However, one cannot rule out that surface-adsorbed viral particles were also considered in the PCR quantification. In mock conditions, none of the samples amplified any significant signal (C_q_ above 34 cycles). To complement the raw data on gene E quantification, a second qPCR targeting the RdRp gene, which codes for RNA-dependent RNA polymerase, was run (Figure 1D). The Spearman correlation test showed a strong relationship between the expression of E and RdRp genes (Spearman r = 0.8545, R-squared = 0.9915, *p* < 0.0001). As TNF-α is one of the most elevated cytokines, with IL-6 and IL-8, in the cerebrospinal fluid of COVID-19 patients [39], the influence of this inflammatory cytokine in modulating the viral entry was investigated using qRT-PCR targeting the viral gene E. The relative amount of gene E was significantly lower when SH-SY5Y cells were exposed to 10 ng/mL of TNF-α concomitantly with the inoculum (one-way ANOVA, 5.6 ± 2.9 vs. 24.0 ± 3.5, *p* < 0.01) (Appendix A). Looking back at the expression of entry receptors under inflammatory condition, we found a significantly decrease in expression of TMPRSS2 and CD147 mRNAs, while ACE2 and NRP1 gene expressions were not statistically different (one-way ANOVA; ACE2, 0.39 ± 0.20 vs. 0.56 ± 0.23; NRP1, 3.1 ± 2.2 vs. 2.10 ± 0.16; *p* > 0.05; TMPRSS2, 8.5 ± 2.1 vs. 1.60 ± 0.41; *p* < 0.05; CD147, 16.00 ± 0.78 vs. 6.60 ± 0.84; *p* < 0.01 for comparison between MOI 5.0 condition and MOI 5.0 + TNF-a) (Appendix A). The viral entry was also assessed by detecting the intracellular expression of nucleocapsid (N) protein in SH-SY5Y cells infected at MOI 1.0 for 24 h. Immunofluorescence showed that N protein was barely found in infected SH-SY5Y and to a much lesser extent than infected VeroE6 cells at same MOI (Figure 1E).

To verify the non-productive infection of SH-SY5Y cells, we used two complementary approaches, while including VeroE6 cells as control permissive cells. First, using an in situ probe hybridizing to the negative-strand RNA (V-nCoV2019-S sense targeting the complementary S gene sequence), the ongoing replication was observed in infected VeroE6 cells at MOI 5.0, while being totally undetectable in infected SH-SY5Y cells at 24 h post-infection (Figure 1F). Second, after thorough washing of the initial inoculum, the supernatants of SH-SY5Y or VeroE6 cells pre-incubated for 2 h to a MOI 1.0, were sampled at different timings (2, 24, 48, and 72 h post-infection). The cell-free medium samples were assessed for their amount of SARS-CoV-2 gene E expression, as this appraisal would indirectly represent the quantity of neo-virions released over time. As shown on Figure 1G and H, the viral inoculum (MOI 1.0) was successfully detected at the time of infection (0 h), regardless of the cell line used, and the expression of gene E in the supernatant was undetectable at two hours of incubation, once the cells were thoroughly washed and fresh medium replaced. While extracellular gene E RNA could not be detected at any time in the refreshed medium of infected SH-SY5Y cells (one-way ANOVA, *p* > 0.05 between 2 h and timings > 2 h) (Figure 1G), a significant amount of gene E RNA was detected in the medium from infected VeroE6 cells. The levels at 48 h post-infection were even higher than the inoculum dose, which demonstrates active replication (one-way ANOVA, *p* < 0.01 between 0 h and 48 h) (Figure 1H).

Complementary and concomitant investigations made with scanning (SEM) and transmission electron microscopy (TEM) were undertaken on mock and infected SH-SY5Y to visualize cell changes between conditions or coronavirus-induced replicative structures (e.g., double-membrane vesicles). No clear specific intracellular changes were recognized as being associated with replicative structures, and certainly no specialized infrastructure containing maturing virions nor release of virions in exocytotic structures were ever found close to the cell surface, and the adjacent extracellular spaces as evidenced in other studies [40]. At this time, one found that all differentiated SH-SY5Y cells revealed large accumulations of lipid droplets (Figure 1I). The lipid droplets from infected SH-SY5Y cells showed an electronic density systematically darker than the one from mock condition (Figure 1J). After gray level analysis of organelles from both conditions, the electron density of lipid droplets from infected cells was significantly higher when normalized to the background and size of each organelle (unpaired *t* test, 42.0 ± 1.6 vs. 20.0 ± 1.4 a.u., *p* < 0.0001) (Figure 1J). Given that several reports pointed out an interplay between lipid droplets, fatty acid metabolism, and the SARS-CoV-2 lifecycle [41,42,43], we investigated the effect of DAPT, a γ-secretase inhibitor known to promote the accumulation of lipid droplets in SH-SY5Y [44]. As expected, the treatment with 10 µM DAPT significantly increased the content of lipid droplets compared to vehicle treated cells (unpaired *t* test, 241.0 ± 6.2 vs. 102.0 ± 1.6%, *p* < 0.0001), but it did not impact the amount of intracellular viral RNA as detected by qPCR against SARS-CoV-2 gene E (unpaired *t* test, 1.00 ± 0.03 vs. 1.00 ± 0.14 fold change, *p* > 0.05) (Figure 1K-L). Of note, no overlap was observed between Oil Red O-stained lipid droplets and the fluorescent dots from the in situ SARS-CoV-2 probe (Figure 1K).

### 3.2. SARS-CoV-2 Induced Morphological Cytopathic Effects on SH-SY5Y Cell Body and Neurites

CPE-related alterations of cell morphology were then studied using light and electron microscopy analysis. Although shortening of neurites was not observed as CPE in infected differentiated SH-SY5Y in our first report, the protocol of differentiation was a bit different to the one used herein [31]. In the present study, SH-SY5Y were differentiated towards neuron-like cells using 10 µM retinoic acid added to medium containing only 1% serum. Accordingly, SH-SY5Y changed their shape and grew neuritic processes but to a shorter extent than with 2% serum (unpaired *t* test, 32 ± 0.62 vs. 51 ± 0.67 µm, *p* < 0.0001) (Appendix A). Mock and infected cells were labeled for cytoskeletal proteins (b3-tubulin and tau protein) (Figure 2A) and neurite length was traced using ImageJ plugin. At 24 h MOI 5.0, SARS-CoV-2 infection provoked a significant decrease in neurite length (unpaired *t* test, 31 ± 0.68 vs. 16 ± 0.53 µm, *p* < 0.0001) (Figure 2B). Neurite shortening refers to the process by which the long, thin extensions of neurons, known as neurites, undergo a reduction in their length, triggered by various cellular stressors, such as oxidative stress or exposure to toxins or viruses. To decipher whether this shortening could be explained by cytoskeletal modifications, the expression levels of microtubule subunits and associated proteins, as well as some of their post-translational modifications, were investigated by immunoblotting experiments. As blots showed, no obvious change in protein expression was seen for tubulin α- and β-subunits, nor for tau protein or some of its phosphorylated epitopes (phospho-tau at Ser202 and Thr205 detected by AT8 antibody; phospho-tau at Ser422), whose phosphorylation promotes tau disassembly and microtubule destabilization (Figure 2C). The semi-quantification of each condition by densitometry supported the unchanged expression of cytoskeletal proteins between mock and infected cells (unpaired *t* test, *p* > 0.05 for all proteins between mock and MOI 5.0) (Figure 2D). The neurite shortening/loss was also noticed at SEM observation (Figure 2E, arrows).

Strikingly, the infected cells showed bleb-like protrusions of the plasma membrane and erupted blebs (Figure 2E, arrowheads). High magnification of infected SH-SY5Y showed these membrane blebs contained extruded cell components of the cytoplasm surrounded by the plasma membrane, whose surface appeared decorated with a pattern of round and barbed protrusions, and the erupted ones revealed lace-like fragile collapsed features (Figure 2F, left). These blebs SEM views were complemented by TEM matching formations (Figure 2F, right) comprising numerous small lipid droplets, membrane parts of endoplasmic reticulum, and numerous proteinaceous granules. The quantification of such figures showed that they were much more frequent in infected cells than in mock cells (unpaired *t* test, 20.0 ± 1.8 vs. 0.4 ± 0.2%, *p* < 0.0001) (Figure 2G). Overall, other SEM observations also highlighted the formation of cell clusters upon SARS-CoV-2 infection at MOI 5.0 (Figure 2E, right panel). The quantification of such clustered cells revealed that they were composed of a larger number of infected cells compared to mock condition (unpaired *t* test, 6.2 ± 0.27 vs. 2.7 ± 0.13 cells/cluster, *p* < 0.0001) (Figure 2H).

CPE-related biochemical alterations in infected SH-SY5Y were assessed using different cell assays. Bearing a dopaminergic phenotype once differentiated, SH-SY5Y cells can release dopamine and neuropeptide Y (NPY) from secretory vesicles into the extracellular medium. Extracellular levels of NPY were quantified using a sensitive sandwich ELISA and were not different between mock and infected SH-SY5Y cells (unpaired *t* test, 198 ± 11 vs. 199 ± 11 pg/mL, *p* < 0.0001) (Figure 2I). Unexpectedly, infected SH-SY5Y showed a slight increase in cell metabolic activity compared to mock condition (unpaired *t* test, 124 ± 8.7 vs. 100 ± 3.6%, *p* < 0.0001) (Figure 2J). Previous data demonstrated low levels of apoptotic SH-SY5Y cells during SARS-CoV-2 infection [31]; those findings were strengthened by the low amount of LDH released in the medium between mock and infected conditions (unpaired *t* test, 1.90 ± 0.17 vs. 1.80 ± 0.13%, *p* < 0.0001) (Figure 2K). However, when TNF-a was added to the medium for 24 h, it induced an obvious cytotoxicity in infected SH-SY5Y cells, which could have negatively impacted their susceptibility to infection (one-way ANOVA, 4.80 ± 0.77 vs. 23.0 ± 2.6%, *p* < 0.001 for comparison between MOI 5.0 condition and MOI 5.0 + TNF-a) (Appendix A).

### 3.3. SARS-CoV-2-Induced Neurite Shortening Was Prevented by TMPRSS2, RBD, and NRP1 Inhibitors

In a final set of experiments, we assessed a panel of four entry inhibitors in the hope of preventing the shortening of neurites, one of the most striking being SARS-CoV-2-induced CPE on SH-SY5Y cells. Confirming previous measurements, SARS-CoV-2 alone at MOI 5.0 induced a significant shortening of neurites over 24 h (one-way ANOVA, 38 ± 0.62 vs. 28 ± 0.49 µm, *p* < 0.0001 for comparison between mock and MOI 5.0) (Figure 3A). The first entry inhibitor was camostat mesylate, an inhibitor of TMPRSS2 protease activity required for the cleavage of the Spike protein and thus essential for infectivity. Upon treatment with camostat mesylate at 0.1 and 10.0 µM, a protection of neurite length was objectivated (one-way ANOVA, 35 ± 1.5 and 39 ± 1.5 µm, *p* < 0.0001 and *p* < 0.0001, for 0.1 and 10.0 µM, respectively) (Figure 3A). A second inhibitor, anti-RBD antibody, binding to the receptor binding domain of the Spike protein, was applied to block its interaction with the host cell ACE2 receptor. Similarly, the incubation with anti-RBD antibody at 0.01 and 1.00 µg/mL prevented the toxicity on neurites induced by SARS-CoV-2 infection (one-way ANOVA, 31 ± 1.3 and 34 ± 1.5 µm, *p* < 0.05 and *p* < 0.0001, for 0.01 and 1.00 µg/mL, respectively) (Figure 3A). A third inhibition strategy relied on the use of human soluble ACE2, a decoy receptor generated via proteolytic cleavage of the membrane anchor. The rationale of its therapeutical indication is a competition with the cognate membrane bound ACE2, thereby siphoning the extracellular pool of infectable virions. None of the concentrations of the recombinant soluble ACE2 exerted any protection against neurite shortening (one-way ANOVA, 32 ± 1.8 and 30 ± 1.9 µm, *p* > 0.05 and *p* > 0.05, for 0.1 and 10.0 µg/mL respectively) (Figure 3A). A fourth inhibitor tested was EG00229, a NRP1 receptor antagonist that selectively inhibits VEGF-A binding and has been involved as co-receptor for SARS-CoV-2 invasion into epithelial cells, endothelial cells, or even central nervous system (CNS) cells [28,29]. EG00229 at 100 µM also significantly protected against neurite shortening (one-way ANOVA, 29 ± 1.2 and 33 ± 1.6 µm, *p* > 0.05 and *p* < 0.001, for 1 and 100 µM respectively) (Figure 3A). To correlate the neuroprotection conferred by the inhibitors with a hypothetical lower amount of internalized viral particles, we assayed the quantity of SARS-CoV-2 gene E mRNA on intracellular fractions. Even though the average amount of gene E transcripts comprised those from the infected-only condition and the mock condition, none of the inhibitor conditions showed a statistically significant decrease, due to insufficient statistical power (one-way ANOVA, *p* > 0.05 for all genes of interest and inhibitor concentrations) (Figure 3B). To circumvent this limitation, for each experimental condition, we computed and correlated, respectively, the average neurite length with the average of intracellular gene E expression. A significant negative correlation, as assessed by Spearman’s rank test, was found between the two variables (Spearman r = −0.8070, R-squared = 0.6512, *p* = 0.0015) (Figure 3C). Finally, the cytotoxicity of each inhibitor concentration was assessed using an LDH assay. While the control condition from lyzed cells released high amount of LDH in the medium (89%), none of the experimental conditions including entry inhibitors showed LDH values greater than 7% (Figure 3D).

## 4. Discussion

Since early 2020, increasing knowledge has been collected about SARS-CoV-2 and its consequences on human health. Symptomatic treatments against respiratory complications or the hyperinflammatory state have rapidly emerged and have allowed the mortality rate observed during the first wave of the pandemic to be reduced. Unfortunately, for some patients, COVID-19 infection still provokes a critical illness with multi-organ involvement, thus requiring hospital care [7,45]. Yet, for other patients, mild to moderate symptoms persist after the resolution of the acute phase and induce long-term disabilities, which is now recognized by the WHO as the long COVID condition [6]. Among the post-acute COVID-19 sequelae, headache, dizziness, cognitive impairments, changes in sensory or pain perception, depression, and anxiety are the most reported neurologic manifestations [2,4,5,6]. Such long-lasting symptoms impact people’s quality of life and remain puzzling for clinicians. Still, the biological basis of the persistent symptoms is unknown, leaving the patients in great uncertainty about their illness or potential recovery. A better understanding of how the virus impacts the CNS is crucial for developing efficient antiviral therapies aimed at alleviating neurological involvements. Cell-based assays are one of the tools available to the scientific community for identifying biologically active compounds against SARS-CoV-2 [15,23,46]. In the present report, SH-SY5Y cells are proven as an easy-to-handle and low-cost tool to study SARS-CoV-2 invasion into neuron-like susceptible cells. They can also serve as phenotypic screening platform for testing CNS-compatible antiviral drugs with a read-out based on SARS-CoV-2-induced CPE.

This study strengthens previous data demonstrating the susceptibility of SH-S5Y5 towards SARS-CoV-2, without leading to a productive infection [24,31,38]. The non-permissive nature of the cells was accompanied by the lack of detection of negative-sense genomic copies, the absence of viral RNA released in the supernatant of cell culture, or lack of evidence of subcellular replicative structures at TEM observation (e.g., double-membrane vesicles, replication membranous web, vacuoles containing virions). Significant expression of common entry genes or facilitators (e.g., ACE2, TMPRSS2, NRP1) supported the fact that SH-SY5Y cells could be specifically targeted by the virus, although the interaction between the receptor-binding domain of Spike and host ACE2 receptor is not always required for invasiveness. Indeed, it has been demonstrated that SARS-CoV-2 can spread from highly permissive cells (VeroE6) to non-permissive cells (SH-SY5Y) through tunneling nanotubes, thereby bypassing the extracellular milieu [38]. The non- or low permissiveness of SH-SY5Y is confirmed by published reports, supporting their conclusion on the lack of detection of viral transcripts in the supernatant of infected cells [24,38,47,48].

Intriguingly, SARS-CoV-2 infection of differentiated SH-SY5Y triggered several markers of cytotoxicity despite no measurable viral genome replication. The absence of viral genome replication does not exclude the expression of non-structural proteins that are translated upon viral entry. For instance, it is well known that Nsp1, one of the first viral proteins being produced, has cytotoxic properties through its inhibition of the translation of the cellular transcripts [49]. The cytotoxicity observed in differentiated SH-SY5Y upon infection could certainly be the result of the expression of one or several non-structural viral proteins. Although SARS-CoV-2 replication cannot be completed in our differentiated neural cells, a productive replication has been demonstrated in several models of brain organoid [50]. Therefore, it is conceivable that high MOI infection occurs in specific areas of the brain.

As severe COVID-19 illness with neurological complications is characterized by neuroinflammation (including perivascular immune infiltrates, local activation of neuroglia, elevated inflammatory markers or cytokines, i.e., IL-6 and TNF-a in the cerebrospinal fluid) [39,51], the additive effect of TNF-a on the SH-SY5Y infection paradigm was tested. Despite the low dose of TNF-a, there was a drastic drop in the internalization of viral RNA. This decrease might be explained either by the concomitant downregulation of viral entry factors (e.g., TMPRSS2 or CD147) or by a significant induction of cell death when SH-SY5Y were sensitized to TNF-a [52]. Complementing previous observations with apoptotic markers, the infection of SH-SY5Y with SARS-CoV-2 alone, even at high MOI, did not induce massive cell lysis [31,47].

Amongst the CPE-related morphological changes, one of the unexpected TEM observations from infected SH-SY5Y was the modification of lipid droplet electron density. Despite the significance not being known and not being investigated by metabolic assays or omics methods, this phenomenon could be induced either specifically by the SARS-CoV-2 or non-specifically following any viral infection. Indeed, in many viral infections, several classes of lipid-derived mediators, including eicosanoids and sphingolipids, are synthetized and regulate the host innate immune response. Based on a described dependency of SARS-CoV-2 towards lipid metabolism [42], we modulated upwards the amount of lipid droplets stored by the cells using DAPT. First, it did not change the amount of internalized viral particles by the SH-SY5Y cells and secondly there was no colocalization between lipid droplets and viral RNA. Our data are thus not in line with others showing that lipid accumulation significantly enhanced the replication of SARS-CoV-2 variants or the ones showing that pharmacological inhibitors of lipid droplet formation hindered SARS-CoV-2 replication [43,53]. One can speculate that this discrepancy relies on a peculiar lipid metabolism in SH-SY5Y cells that would confer a non-permissive character to the virus. Other infections, like Zika virus, are known to reprogram the cell metabolism and impact lipid droplet morphology or content [54]. As suggested by the slight increase in overall cellular metabolic activity (MTT assay), SARS-CoV-2 can mobilize host neuron machinery to its advantage, even though, some intrinsic defects in SH-SY5Y cells might make the virus unable to reach replicative steps. Such a hypermetabolic state was shown in other cell lines such as HuH7 cells as well as in brain organoids infected by SARS-CoV-2 [8,24].

Fine morphological alterations in infected SH-SY5Y also included cell aggregation, formation of bleb-like protrusions at the plasma membrane and neurite shortening. These cytopathic changes recall abnormal cell adhesion, virus-induced syncytium formation, and/or disturbances of the cytoskeleton because of the infection. If fusion between cells making syncytia were not a surprise among tumor cells such as neuroblastoma cells, the increased SH-SY5Y aggregates concurrently confirmed infectivity as Spike protein-induced cell fusions occurred as in other SARS-CoV-2-infected cell lines [55,56]. As established in the literature, blebs with intracellular components (lipid droplets, granular proteinaceous content, endoplasmic reticulum, cytoskeletal elements) are not necessary signs of apoptotic or necrotic processes. Instead, it can reflect either incomplete formation of motility appendages, aborted due to disturbed associated cytoskeletal components during the infection [57,58,59], or another interpretation can be the expulsion of vesicles formed or precursors for the components of virions, but curtailed.

In addition, neurite shortening was another morphological change following SARS-CoV-2 infection of SH-SY5Y cells. Even though we tried to explain such a cytopathic effect via cytoskeletal alterations, no changes in the expression of microtubule subunits or microtubule-associated proteins (e.g., tau protein) were objectivated. Conversely, a transcriptomic study on neuroblastoma SK-N-SH cells revealed that SARS-CoV-2 infection induced a downregulation of several members of building blocks of the cytoskeleton, as well as some of their regulators (i.e., *TUBA1A, TUBA4A, TUBB4A, STMN4, TMSB15A, SYNPO2*) [60]. In brain organoids, the infection with SARS-CoV-2 induced a relocation of tau protein from neurite to neuronal cell soma and was associated with tau hyperphosphorylation at Thr231 [46]. Abnormal post-translational modifications of tau are known to promote its disassembly from microtubules and destabilize the microtubule network, such as that described in tauopathies. Changes in localization, in protein expression or in hyperphosphorylations of tau protein (Ser202/Thr205, Ser422) were not evidenced in our infected SH-SY5Y cells, although all the phospho-epitopes were not exhaustively investigated. Conversely, an available preprint suggests that tau hyperphosphorylation on epitopes Ser262 and Ser396, occurs in SARS-CoV-2-infected SH-SY5Y cells, and even leads to tau aggregation [61]. If verified, such alterations of cytoskeletal proteins and destabilization of neuritic architecture may be the biological basis of impairments in neuronal functions. The SARS-CoV-2 Spike protein, per se, seems to act as a toxic factor towards fine neuronal processes by inducing local endolysosome dysfunction and neuritic varicosities [62]. A recent finding relating to neurite shortening and cell clustering is the discovery of the fusogenic role of the Spike protein, able to cause neurite–neurite, neuron–neuron or even neuron–glia fusion, at the origin of the formation of multicellular syncytia in infected brain organoids [63]. Further research is needed to decipher the molecular basis of SARS-CoV-2 neuritotoxicity in non-permissive neuronal cells: e.g., does it rely on the early expression of non-structural proteins such as nsp1, endowed with cytotoxic features? Does the neurite shortening further compromise synaptic contacts, the release of neurotransmitters, and electrical activity in more complex and relevant neuronal networks?

FDA- or EMA-approved molecules are already in use and have demonstrated their efficiency to block SARS-CoV-2 entry or replication and therefore limit viral spread [64]. Many of them are monoclonal antibodies that target the Spike protein (tixagevimab, cilgavimab, sotrovimab, regdanvimab, casirivimab, imdevimab, etc.). In the present infection paradigm, given the lack of SARS-CoV-2 replication in SH-SY5Y, the only relevant pharmacological approach to hinder the cytopathic effect was to limit the viral entry. Hence, we sought to evaluate the abilities of four inhibitors to limit the entry into the neuron-like cells and to protect against SARS-CoV-2-induced neurite shortening. Neurite shortening was used as a relevant and easy-access read-out for assessing the neurocytopathy, despite the absolute neurite length in differentiated SH-SY5Y culture could be subjected to batch-to-batch variabilities upon experimental conditions. This is why we systematically recommend to include a non-treated/vehicle condition in *each* run, and for the measurements to be performed by independent blind investigators, or better, via automated image analysis. On these bases, three candidate inhibitors, camostat mesylate, anti-RBD antibody, and EG00229—blocking, respectively, TMPRSS2 activity, Spike protein, or NRP1 coreceptor—demonstrated a significant protection on neurites. This suggests that SARS-CoV-2 relies on TMPRSS2-dependent processing, RBD binding, and NRP1 binding before invading SH-SY5Y cells, even though this conclusion should deserve caution in our conditions and further investigations with, e.g., loss-of-function experiments of entry genes. As shown by other groups, entry of SARS-CoV-2 into neurons greatly depends on ACE2-RBD binding, but also on TMPRSS2 activity and NRP1 interaction [8,29,30]. One must mention that we were not able to appreciate the direct viral entry inhibition, as assessed by the quantity of intracellular viral RNA due to an insufficient statistical power (small number of samples). However, encouraging data came from the analysis of a negative correlation between the amount of intracellular viral RNA and overall neurite length, as well as the low toxicity of all the inhibitors tested. For an unknown reason, human recombinant soluble ACE2 did not rescue the neurite changes caused by SARS-CoV-2 toxicity, whereas this compound had showed efficient entry inhibition both on kidney organoids and in patients [65,66].

In conclusion, we herein characterized a cell-based assay to better understand the neuro-invasion and alleviate the virulence of SARS-CoV-2 towards neuron-like cells. Various morphological SARS-CoV-2-induced cytopathic effects were highlighted, among which the most easily translatable for high-throughput compound screening are the neurite shortening effect and the cell cluster formation. Although our report is limited to the study of the neurocytopathic effect of pre-alpha SARS-CoV-2 strain, we are convinced that it can be applied to all circulating variants with success according to the neurotropism of the variant of concern. Our results indicate that SARS-CoV-2 infection into non-permissive neuron-like cells can initiate morphological changes in neuronal cell bodies and neuritic network, putatively causing alteration in neuronal communication—in the absence of neuronal death—and revealing a possible pathomechanism underlying COVID-19 neurological complications. The main findings were that SH-SY5Y cells displayed susceptibility to SARS-CoV-2 infection without replicative abilities; however, following infection, these cells showed alterations in cell morphology (blebs at the plasma membrane, cell aggregation, neurite shortening); and the cytopathic effect on neurites could be reversed in presence of camostat mesylate, anti-RBD neutralizing antibody, and NRP1 inhibitor.

## Figures and Tables

**Figure 1 viruses-15-02020-f001:**
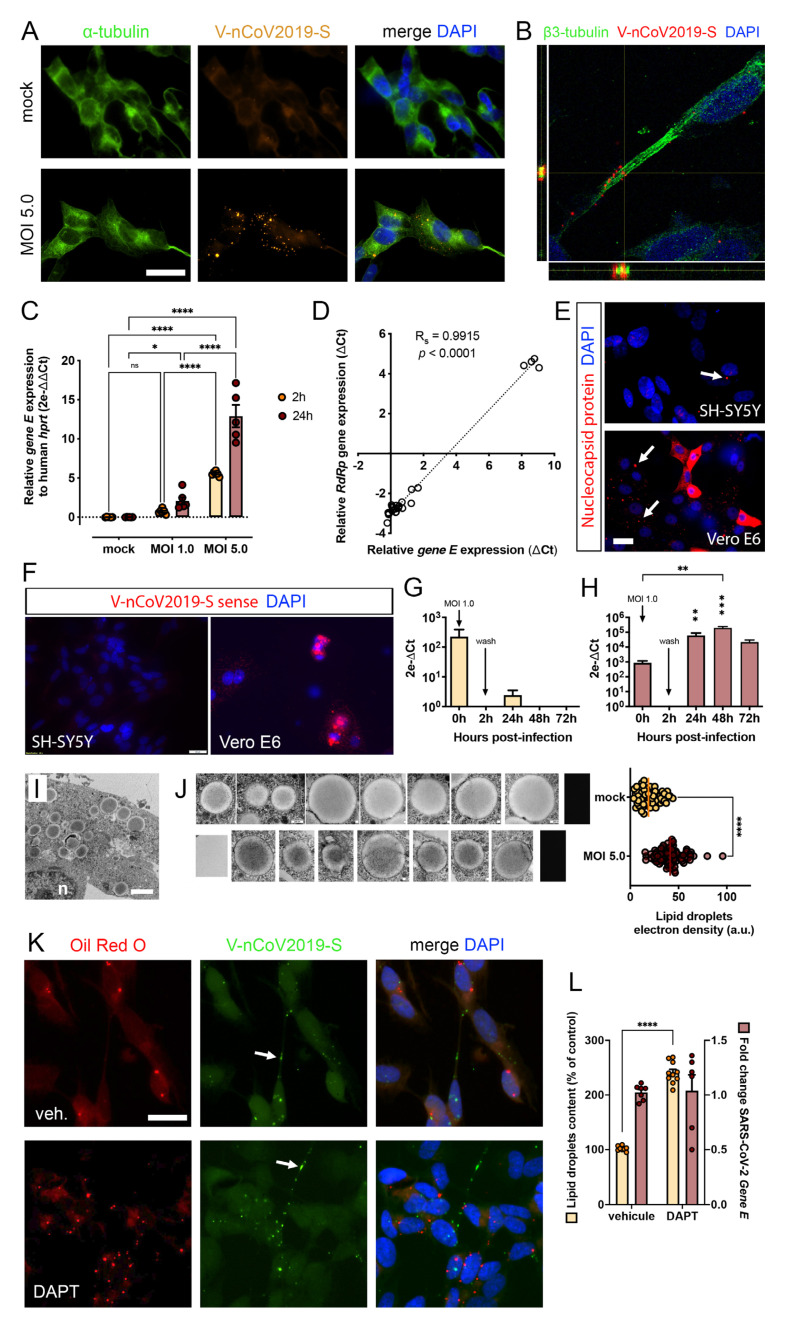
Susceptibility of differentiated SH-SY5Y towards SARS-CoV-2. Fluorescent in situ (*ish*) hybridization using SARS-CoV-2 gene S probe (V-nCoV2019-S, red) on SH-SY5Y cells exposed to SARS-CoV-2 at MOI 5.0 during 24 h or to mock condition (**A**). The cytoskeleton was immunostained using anti-α-tubulin antibody (green). Nuclei were counterstained with DAPI (blue). Optical slice through infected SH-SY5Y cell obtained by confocal microscopy (**B**). The cytoskeleton was immunostained using anti-β3-tubulin antibody (green) and viral RNA using the V-nCoV2019-S *ish* probe (red). Nuclei were counterstained with DAPI (blue). Quantification of SARS-CoV-2 gene E relative abundance in infected SH-SY5Y cells according to various MOIs and timings of infection (**C**). Ct cycles for *gene E* were normalized to Ct cycles of human *hprt*, and the relative expression data were expressed as 2e-ΔCt. Statistical analysis were computed using two-way ANOVA following Tukey multiple comparison test. n = 5–6 biological replicates for each condition. The degree of correlation between the relative expression of *RdRp* gene and of the *E* gene was computed using a Spearman rank test (R-squared = 0.9915, *p* < 0.0001) (**D**). The SARS-CoV-2 Nucleocapsid protein was immunostained using anti-SARS coronavirus nucleocapsid antibody (red). Nuclei were counterstained with DAPI (blue) (**E**). In many VeroE6 cells, the cytoplasm was fully loaded with viral proteins, while in SH-SY5Y cells, the viral protein N was barely detected (arrows). Fluorescent *ish* against the negative-strand RNA using anti-sense SARS-CoV-2 gene S probe (V-nCoV2019-S-sense, red) on SH-SY5Y cells and VeroE6 cells exposed to SARS-CoV-2 at MOI 5.0 for 24 h (**F**). Nuclei were counterstained with DAPI (blue). Quantification of SARS-CoV-2 gene E relative abundance in the supernatant of SH-SY5Y (**G**) and VeroE6 (**H**) cells at various timings post-infection. Cells were infected with an MOI 1.0. for 2 h and then thoroughly washed 3 times before a virus-free medium was refreshed. Ct cycles for *gene E* were normalized to Ct cycles for *gene E* at 2 h post-infection. Data are expressed as 2e-ΔCt. Statistical analysis were computed using one-way ANOVA followed by Dunn multiple comparison test. n = 3 biological replicates for SH-SY5Y and n = 6 biological replicates for VeroE6. Representative TEM illustrations of infected SH-SY5Y (**I**) and their lipid droplets (**J**). Lipid droplets electron densities from mock (upper panels) and infected (lower panels) conditions were assessed via gray level analysis (**J**). Statistical analysis was computed using unpaired *t* test, including at least 59 droplets for each condition. Intracellular lipid droplets and viral RNA were stained, respectively, using Oil Red O (red) and fluorescent *ish* probe (V-nCoV2019-S, green). Nuclei were counterstained with DAPI (blue) (**K**). Lipid droplets and viral RNA were quantified, respectively, via image analysis and intracellular gene *E* expression (**L**). n = at least 6 biological replicates for each condition. The data on plots are represented as mean ± SEM. *p* = ns, non-significant; * *p* < 0.05; ** *p* < 0.01; *** *p* < 0.001; **** *p* < 0.0001. Scale bars represent 20 µm in (**A**,**E**,**F**,**K**). Scale bar represents 1 µm in (**I**).

**Figure 2 viruses-15-02020-f002:**
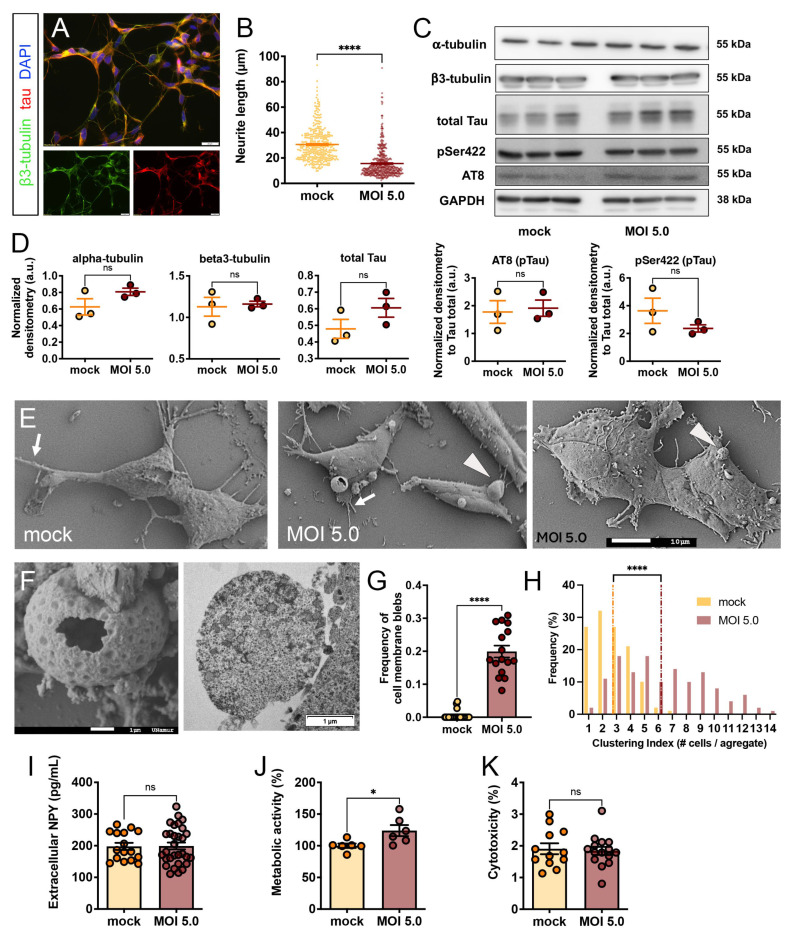
SARS-CoV-2-induced cytopathic effects on SH-SY5Y cells. The neuritic network was immunostained using anti-β3-tubulin antibody green) and tau antibody (red) (**A**) Nuclei were counterstained with DAPI (blue). Neurite length was measured using ImageJ tool in mock SH-SY5Y and infected SH-SY5Y at MOI 5.0 for 24 h (**B**). Statistical analysis was computed using unpaired *t* test, including at least 372 neurite traces for each condition. The expressions of cytoskeletal proteins (a- and b-tubulin), total tau protein as well as phosphorylated tau epitopes (AT8, pSer422) were assessed on immunoblots. GAPDH was used as loading control (**C**). Semi-quantification of pixel density from immunoblots (**D**). The expression of protein of interest was related to the loading control, except phospho-tau epitopes that were normalized on total tau levels. Statistical analysis was computed using Mann–Whitney *t* test. n = 3 biological replicates for each condition. Representative SEM illustrations of mock SH-SY5Y (left panel) and infected SH-SY5Y (middle and right panels) (**E**). Arrows indicate neuritic processes, and arrowheads indicate the bleb-like protrusions. High magnification of bleb-like protrusions obtained by SEM (left) and TEM (right) (**F**). The images are representative of one experiment out of two independent experiments. On SEM images, the frequency of cell membrane blebs per field of view was quantified in mock SH-SY5Y and infected SH-SY5Y at MOI 5.0 for 24 h (**G**). At least 16 fields of view were investigated per condition. Statistical analysis was computed using unpaired *t* test. On SEM images, the cell clustering was quantified in both experimental conditions (**H**). The average of each distribution is shown on the histograms using dashed lines (yellow for mock; red for MOI 5.0). The amount of neuropeptide Y (NPY) released in the culture supernatant was assessed by ELISA in both experimental conditions (**I**). MTT assay was used to assess the NADPH-dependent cell metabolic activity (**J**)**.** Data were expressed in % compared to the basal level of mock SH-SY5Y. The amount of lactate deshydrogenase (LDH) released in the culture supernatant was assessed by LDH assay, and data were expressed in % compared to the maximal amount of LDH released by totally lyzed cells (**K**). The data on plots are represented as mean ± SEM. *p* = ns, non-significant; * *p* < 0.05 **** *p* < 0.0001. Scale bars represent 20 µm, 10 µm, and 1 µm, respectively, in (**A**,**E**,**F**).

**Figure 3 viruses-15-02020-f003:**
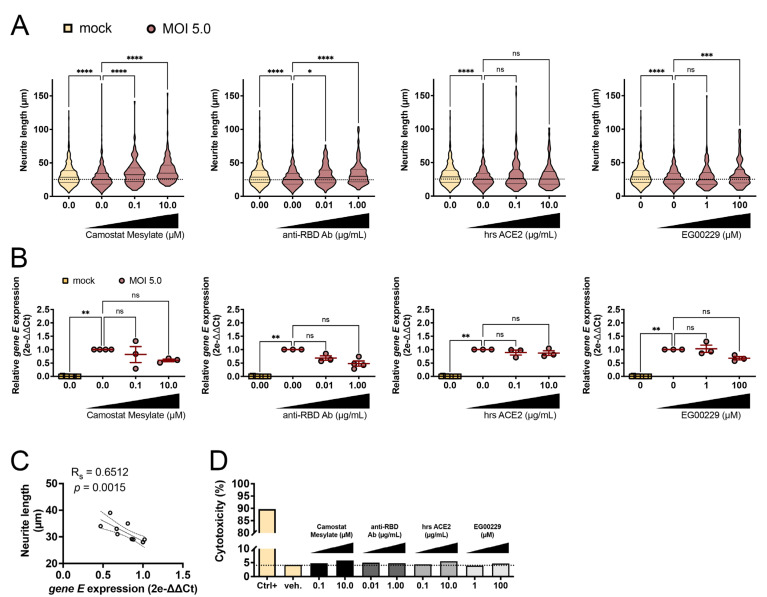
Effect of viral entry inhibitors on SARS-CoV-2-induced neurite shortening and intracellular viral RNA. Camostat mesylate, anti-RBD antibody, human recombinant soluble ACE2, or EG00229 were incubated at increasing concentrations with the viral inoculum, MOI 5.0, for 24 h. Neurite length was measured using ImageJ tool in mock SH-SY5Y, infected SH-SY5Y, and infected SH-SY5Y plus entry inhibitors (**A**). Statistical analysis was computed using one-way ANOVA followed by Dunn multiple comparison test, compared to the infected SH-SY5Y (w/o inhibitor). Data are represented as violin plots from one out of two independent experiments. Quantification of intracellular SARS-CoV-2 gene *E* relative abundance in mock SH-SY5Y, infected SH-SY5Y and infected SH-SY5Y plus entry inhibitors (**B**). Ct cycles for gene *E* were normalized to Ct cycles of human *hprt*, and the relative expression data were expressed as 2e-ΔΔCt. Statistical analysis was computed using one-way ANOVA followed by Dunn multiple comparison test, compared to the infected SH-SY5Y (w/o inhibitor). n = 3 biological replicates for each condition. The degree of correlation between the relative expression of *E* gene and neurite length across all the experimental conditions was computed using a Spearman rank test (R-squared = 0.6512, *p* = 0.0015) (**C**). The amount of lactate deshydrogenase (LDH) released in the supernatant of cells exposed to the various entry inhibitors was assessed by LDH assay (**D**)**.** Ctrl+ and veh. conditions correspond, respectively, to SH-SY5Y exposed to DMSO 20% (positive control) and to DMSO 0.1% (vehicle-only negative control). The data on plots are represented as mean ± SEM. *p* = ns, non-significant; * *p* < 0.05; ** *p* < 0.01; *** *p* < 0.001; **** *p* < 0.0001.

**Table 1 viruses-15-02020-t001:** Primer and probe sequences for RT-qPCR.

*Gene Name*	Primer Sense 5′-3′	Primer Antisense 5′-3′
SARS-CoV-2 *gene E*	5′-ACAGGTACGTTAATAGTTAATAGCGT-3′	5′-ATATTGCAGCAGTACGCACACA-3′
	Taqman probe: (FAM)-ACACTAGCCATCCTTACTGCGCTTCG-(BHQ1)	
SARS-CoV-2 *gene RdRp*	5′-AGAATAGAGCTCGCACCGTA-3′	5′-CTCCTCTAGTGGCGGCTATT-3′
Human *hprt*	5′-TGACACTGGCAAAACAATGCA-3′	5′-GGTCCTTTTCACCAGCAAGCT-3′
Human *ace2*	5′-GTGCACAAAGGTGACAATGG-3′	5′-GGCTGCAGAAAGTGACATGA-3′
Human *tmprss2*	5′-CACTGTGCA TCACCTTGACC-3′	5′-ACACGCCATCACACCAGTTA-3′
Human *cd147*	5′-ATGCTGGTCTGCAAGTCAGA-3′	5′-GCGAGGAACTCACGAAGAAC-3′
Human *nrp1*	5′-GAAGCACCGAGAGAACAAGG-3′	5′-AGTCCGCAGCTCAGGTGTAT-3′

**Table 2 viruses-15-02020-t002:** Primary antibodies used for Western Blotting and Immunofluorescence.

Antibody Name	Species	Reference	Application	Dilution
Anti-βIII-tubulin	Mouse	Ab78078, Abcam	Western Blotting Immunofluorescence	1:5000 1:500
Anti-α-tubulin	Mouse	T5168, Sigma-Aldrich	Western Blotting Immunofluorescence	1:1000 1:500
Anti-Tau total	Rabbit	A0024, Dako	Western Blotting Immunofluorescence	1:5000 1:100
Anti phospho-Tau (Ser202, Thr205) AT8	Mouse	MN1020, Invitrogen	Western Blotting	1:1000
Anti-phospho-Tau (Ser422)	Mouse	4BDX-1501, 4BioDX	Western Blotting	1:5000
Anti-GAPDH	Mouse	G8795-100 UL, Sigma-Aldrich	Western Blotting	1:10,000
SARS Coronavirus Nucleocapsid	Rabbit	PA1-41098, Invitrogen	Immunofluorescence	1:100

**Table 3 viruses-15-02020-t003:** Secondary antibodies used for Western Blotting and Immunofluorescence.

Antibody Name	Reference	Application	Dilution
Anti-Mouse IgG, HRP-Linked antibody	7076S, Cell Signaling	Western Blotting	1:1000
Anti-Rabbit IgG, HRP-Linked antibody	7074S, Cell signaling	Western Blotting	1:1000
Alexa fluor 488 Goat anti-mouse	A11001, Invitrogen	Immunofluorescence	1:200
Alexa fluor 568 Goat anti-rabbit	A11011, Life Technologies	Immunofluorescence	1:200

## Data Availability

The data presented in this study (original images from light and electron microscopy, uncropped blots, raw data and statistics, etc.) are openly available in PURE database of University of Namur at https://pure.unamur.be/admin/files/86482430/Mignolet_et_al_-_Viruses_-_Data_Open_Availability.zip (accessed on 22 September 2023).

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
