# Peer review of "Viral Entry Inhibitors Protect against SARS-CoV-2-Induced Neurite Shortening in Differentiated SH-SY5Y Cells"

_viruses, 2023, doi:10.3390/v15102020_

Round 1

Reviewer 1 Report

This study found that the susceptibility of SH-SY5Y to SARS-CoV-2 was confirmed at high multiplicity of-infection, without viral replication or release. Infection caused a reduction of the length of neuriticprocesses, occurrence of plasma membrane blebs. And infection caused a reduction of the length of neuritic processes, occurrence of plasma membrane blebs, cell clustering and changes in lipid droplets electron density. At last the authors demonstrated that the neurite shortening could be prevented by the highest concentration of camostat mesylate, anti-RBD antibody, and NRP1 inhibitor, but not by soluble ACE2. They concluded that targeting specific SARS-CoV-2 host receptors could reverse its neurocyto-pathic effect on SH-SY5Y.

Yes, this study found some characteristics of SARS-CoV-2 infection to neuroblastoma cell lines SH-SY5Y, but it is not clear about the relationship between them. And the possible reason of that viral entry inhibitors protect against SARS-CoV-2-induced 2 neurite shortening in differentiated SH-SY5Y cells is not mentioned. So it is necessary to perform more experiments to confirm the conclusion and the mechanisms.

Some suggestion:

1. In the results part, to easy understand the results of figure, the authors should give a conclusion for each figure.

2. The authors described that SH-SY5Y is non-permissive cells, it is necessary to discuss by combining other reports.

3. It is not clear that the logical relationship between the upper box and bottom box of graphical abstract.

Reviewer 2 Report

The study carried out by Mignolet et al, is clear and provides a succinct summary of the research focus. However, it could benefit from some minor improvements:

 -Considering highlighting the specific inhibitor used in the study in the title, would help to provide more context and will make the title more informative.

-While neurite shortening is clear to scientists, the term might not be understood by a broader audience. A brief description of 2-3 lines can make it more accessible.

-While the authors are confident that their approach can be applied to all circulating variants; however, missed to emphasize how this research impacts our understanding of SARS-CoV-2 and its effects on the nervous system in the context of ongoing public health concerns.

-Are there any specific aspects or questions that remain unexplored and could be addressed in follow-up studies?—should be discussed.

The study carried out by Mignolet et al, is clear and provides a succinct summary of the research focus. However, it could benefit from some minor improvements:

 -Considering highlighting the specific inhibitor used in the study in the title, would help to provide more context and will make the title more informative.

-While neurite shortening is clear to scientists, the term might not be understood by a broader audience. A brief description of 2-3 lines can make it more accessible.

-While the authors are confident that their approach can be applied to all circulating variants; however, missed to emphasize how this research impacts our understanding of SARS-CoV-2 and its effects on the nervous system in the context of ongoing public health concerns.

-Are there any specific aspects or questions that remain unexplored and could be addressed in follow-up studies?—should be discussed.

Reviewer 3 Report

This study demonstrated that targeting specific host receptors of SARS-CoV-2 can reverse its neurocytopathic effects on SH-SY5Y cells. These findings contribute to our understanding of the cellular responses to SARS-CoV-2 infection in neuronal cells and provide insights into potential therapeutic strategies for preventing neurocytopathic effects caused by the virus.

I recommend accepting this article after MINOR REVISIONS.

1. This study demonstrates that targeting specific host receptors of SARS-CoV-2 can reverse its neurocytopathic effects on SH-SY5Y cells. By using entry inhibitors, such as camostat mesylate, anti-RBD antibody, and NRP1 inhibitor, the researchers were able to prevent neurite shortening induced by the pre-alpha strain of SARS-CoV-2. Further research is needed to explore the effectiveness of these entry inhibitors on other SARS-CoV-2 variants and their potential application in mitigating neurocytopathic effects in vivo.

2. For the benefits of the readers, in lines 53-54 (References [10-15]), please supply more relevant knowledge.

3. There is a lack of recent literature citations. The authors should enrich the related articles published between 2020 and 2023. For example, in lines 38-41, “Although considered as a respiratory pathogen, SARS-CoV-2 (Severe Acute Respiratory Syndrome-CoronaVirus-2), the virus that caused the coronavirus disease (COVID) 2019 pandemic, is also responsible in a substantial number of cases of extrapulmonary manifestations.(Nutrients 2023, 15, 3443)”; in lines 55-57, “Independent research groups have shown in vitro that several neuronal populations support the entry of SARS-CoV-2, followed or not by genome replication, production, and release of virions. (?)”; in lines 67-69, “To enter the host neurons, SARS-CoV-2 requires the interaction between its Spike (S) protein and cell membrane receptors. This Spike protein is a glycoprotein containing two subunits called S1 and S2. (?)”.

4. “COVID” please change with “COVID-19”; “coronavirus disease (COVID) 2019” please change with “COVID-19”.

5. The quality of the “4. Discussion” must be improved.

Minor editing

Round 2

Reviewer 1 Report

The authors has made more revisions on the manuscript, I think you can do more investigation on the mechanisms of your fidings in the next research. This manuscript can be publish in present form.